# CLIP-It!
# Language-Guided Video Summarization

**Medhini Narasimhan      Anna Rohrbach      Trevor Darrell**
University of California, Berkeley
{medhini, anna.rohrbach, trevordarrell}@berkeley.edu
https://medhini.github.io/clip_it

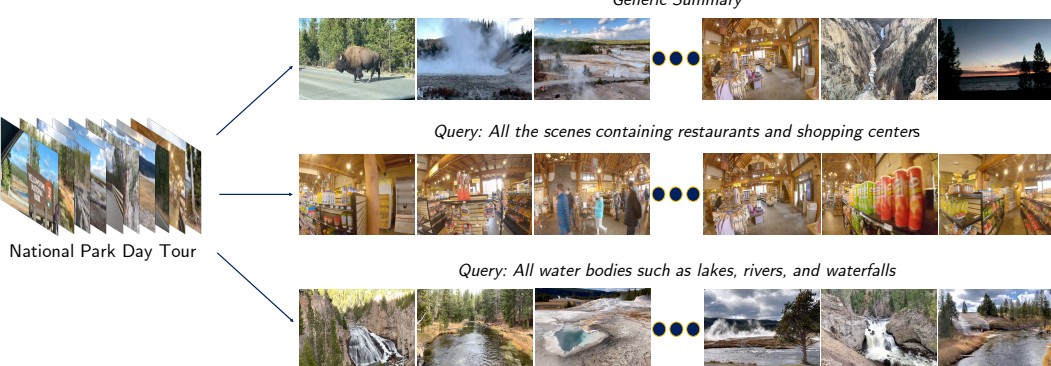

Figure 1: We introduce CLIP-It, a *language-guided* multimodal transformer for generic and query-focused video summarization. The figure shows results from our method. Given a day-long video of a national park tour, the generic summary (top) is a video with relevant and diverse keyframes. When using the query "All the scenes containing restaurants and shopping centers", the generated query-focused summary includes all the matching scenes. Similarly, the query "All water bodies such as lakes, rivers, and waterfalls", yields a short summary containing all the water bodies present in the video.

## Abstract

A generic video summary is an abridged version of a video that conveys the whole story and features the most important scenes. Yet the importance of scenes in a video is often subjective, and users should have the option of customizing the summary by using natural language to specify what is important to them. Further, existing models for fully automatic generic summarization have not exploited available language models, which can serve as an effective prior for saliency. This work introduces *CLIP-It*, a single framework for addressing both generic and query-focused video summarization, typically approached separately in the literature. We propose a *language-guided* multimodal transformer that learns to score frames in a video based on their importance relative to one another and their correlation with a user-defined query (for query-focused summarization) or an automatically generated dense video caption (for generic video summarization). Our model can be extended to the unsupervised setting by training without ground-truth supervision. We outperform baselines and prior work by a significant margin on both standard video summarization datasets (TVSum and SumMe) and a query-focused video summarization dataset (QFVS). Particularly, we achieve large improvements in the transfer setting, attesting to our method's strong generalization capabilities.

## 1   Introduction

An effective video summary captures the essence of the video and provides a quick overview as an alternative to viewing the whole video; it should be succinct yet representative of the entire video. Summarizing videos has many use cases - for example, viewers on YouTube may want to watch a

35th Conference on Neural Information Processing Systems (NeurIPS 2021).

short summary of the video to assess its relevance. While a generic summary is useful for capturing the important scenes in a video, it is even more practical if the summary can be customized by the user. As seen in Fig. 1, users should be able to indicate the concepts of the video they would like to see in the summary using natural language queries.

Generic video summarization datasets such as TVSum [36] and SumMe [7] provide ground-truth annotations in the form of frame or shot-level importance scores specified by multiple annotators. Several learning-based methods reduce the task to a frame-wise score prediction problem. Sequence labeling methods [5, 42, 43, 44, 18] model variable-range dependencies between frames but fail to capture relationships across all frames simultaneously. While attention [3] and graph based [26] methods address this partly, they disregard ordering of the input frames which is useful when predicting scores. Moreover, these methods use only visual cues to produce the summaries and cannot be customized with a natural language input. Another line of work, query-focused video summarization [32], allows the users to customize the summary by specifying a query containing two concepts (eg., food and drinks). However, in their work the query can only be chosen from a fixed set of predefined concepts which limits the flexibility for the user.

It is important to note that efforts in generic and query-focused video summarization have so far been disjoint, with no single method for both. Our key innovation is to unify these tasks in one *language-guided* framework. We introduce *CLIP-It*, a multimodal summarization model which takes two inputs, a video and a natural language text, and synthesizes a summary video conditioned on the text. In case of generic video summarization, the natural language text is a video description obtained using an off-the-shelf dense video captioning method. Alternatively, in the case of query-focused video summarization, the language input is a user-defined query. Unlike existing generic methods which only use visual cues, we show that adding language as an input leads to the "discovery" of relevant concepts and actions resulting in better summaries. Our method uses a Transformer with positional encoding, which can attend to all frames at once (unlike LSTM based methods) and also keep track of the ordering of frames (unlike graph based methods). In contrast to existing query-focused methods [32] which only allow keyword based queries, we aim to enable open-ended natural language queries for users to customize video summaries. For example, as seen in Fig. 1, using our method users can specify long and descriptive queries such as, "All water bodies such as lakes, rivers, and waterfalls" which is not possible with previous methods.

Given an input video, CLIP-It generates a video summary guided by either a user-defined natural language query or a system generated description. It uses a *Language-Guided Attention* head to compute a joint representation of the image and language embeddings, and a *Frame-Scoring Transformer* to assign scores to individual frames in the video using the fused representations. Following [43, 42], the summary video is constructed from high scoring frames by converting frame-level scores to shot-level scores and using knapsack algorithm to fit the maximum number of high scoring shots in a timed window. Fig. 1 shows an output from our method. Given an hour long video of a national park tour, we generate a 2 minute generic summary comprising of all the important scenes in the video. Given the two language queries, our method picks the matching keyframes in both cases. We can train CLIP-It without ground-truth supervision by leveraging the reconstruction and diversity losses [9, 31]. For generic video summarization, we evaluate our approach on the standard benchmarks, TVSum and SumMe. We achieve performance improvement on F1 score of nearly 3% in the supervised setting and 4% in the unsupervised setting on both datasets. We show large gains (5%) in the Transfer setting, where CLIP-It is trained and evaluated on disjoint sets of data. For the query-focused scenario we evaluate on the QFVS dataset [33], where we also achieve state-of-the-art results.

To summarize our contributions, we introduce CLIP-It, a *language-guided* model that unifies generic and query-focused video summarization. Our approach uses language conditioning in the form of off-the-shelf video descriptions (for generic summarization) or user-defined natural language queries (for query-focused summarization). We show that the Transformer design enables effective contextualization across frames, benefiting our tasks. We also demonstrate the impact of language guidance on generic summarization. Finally, we establish the new state-of-the-art on both generic and query-focused datasets in supervised and unsupervised settings.

## 2 Related Work

**Generic Video Summarization.** A video summary is a short synopsis created by stitching together important clips from the original video. Early works [21, 35, 34] referred to it as Video Skimming or Dynamic Video Summarization and used hand-designed features to generate summaries. Likewise,

non-parametric unsupervised methods [13, 16, 23, 19, 20, 28] used various heuristics to represent the importance of frames. Introduction of benchmark datasets such as TVSum [36] and SumMe [7] provided relevance scores for frames in videos annotated by users, resulting in multiple human generated summaries. This enabled automatic evaluation of video summarization techniques and has lead to the development of many supervised learning based methods [6, 8, 22, 26, 31, 30, 41, 42, 43, 44, 45]. These approaches capture high-level semantic information and outperform the heuristic unsupervised methods. Fully convolutional sequence networks [31] treat video summarization as a binary label prediction problem. Determinantal point processes [15] and LSTM [10] based approaches [5, 18, 42, 43, 44] model variable-range dependencies between frames. However, these are sequential and fail to capture relationships across all frames simultaneously. Attention [3] and graph based methods [26] address this issue by modeling relationships across all frames, but they disregard the ordering of frames in a video, which is also useful for summarization. Our method uses a Transformer [38] with positional encoding, which allows for joint attention across all frames while maintaining an ordering of the input sequence of frames.

Some of the above works have supervised and unsupervised variants with modifications to the objective function. Specifically, for the unsupervised variant, they use reconstruction and diversity losses which do not require ground truth. We follow prior work in terms of using the same loss functions. Other notable unsupervised approaches include an adversarial LSTM based method [22], a generative adversarial network to learn from unpaired data [30], and a cycle consistent learning objective [41].

**Video-Text Summarization.** Plummer *et al*. [27] use image-language embeddings for generating video summaries but evaluate their approach in the text domain, and not on the generic video summarization benchmarks. Furthermore, they require text to be provided as input and don't have a mechanism to generate captions if absent. On the other hand, our method works well with off-the-shelf captions and we evaluate on both generic and query-focused benchmark datasets. Chen *et al*. [1] jointly train a text and video summarization network but rely on ground-truth text summaries.

**Query-Focused Video Summarization.** Oftentimes users browsing videos on YouTube are looking for something specific so a generic summary might not suffice. In this case, there should be an option to customize the generated summary using a natural language query. Sharghi *et al*. [32] introduce the Query-Focused Video Summarization (QFVS) dataset for UT Egocentric [16] videos containing user-defined video summaries for a set of pre-defined concepts. Sharghi *et al*. [33] propose a memory network to attend over different video frames and shots with the user query as input. However, this recurrent attention mechanism precludes parallelization and limits modeling long-range dependencies, which is overcome by our Transformer architecture. Moreover, their method only works with the pre-defined set of keyword based queries in QFVS dataset. Since we use CLIP [29] to encode the language input and train our method on dense video descriptions, this allows users to define freeform queries at test time (as seen in Fig. 1). Other works [12, 39] similarly condition the summary generation on keyword based queries but haven't released their data.

**Transformers.** Transformers [38] were introduced for neural machine translation and have since been applied to video-language tasks such as video-retrieval [4] and video captioning [17, 48]. In this work we adapt transformers for video summarization. We modify self-attention [38] to a Language-Guided Attention block that accepts inputs from two modalities. Additionally, our method also relies on CLIP [29] for extracting image and text features and a Bi-Modal Transformer [11] for dense video caption generation, both of which also have transformer backbone.

## 3 CLIP-It: Language-Guided Video Summarization

CLIP-It is a unified *language-guided* framework for both generic and query-focused video summarization. It takes an input video and a user-defined query or a system generated dense video caption and constructs a summary by selecting key frames from the video. First, we explain the intuition behind our approach. In the case of query-focused summarization, clearly, it is necessary to model the user query as input in order to produce an appropriate summary. In the case of generic video summarization no user query is available; nonetheless, we show here that we can leverage the semantic information contained in associated natural language descriptions to guide the summarization process. Assuming we had a generic description that accompanies a video (e.g., *A person is walking a dog. The person throws a ball. The dog runs after it.*), we could leverage its semantic embedding to *match it* to the most relevant portions of the video. Such a description could be obtained automatically by generating dense video captions [11]. We first present an overview of our approach, CLIP-It, followed by a detailed description of the individual components.

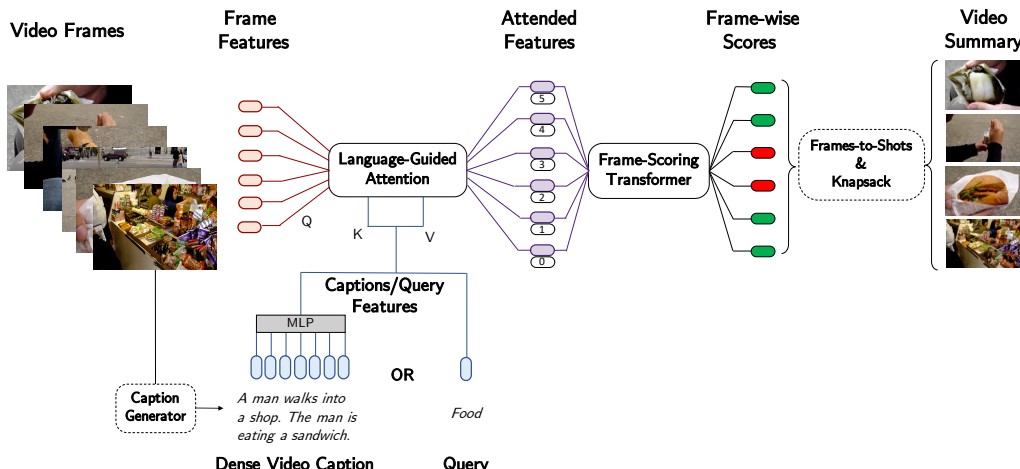

Figure 2: **Overview of CLIP-It.** Given an input video, CLIP-It generates a summary conditioned on either a user-defined natural language query or an automatically generated dense video caption. The Language-Guided Attention head fuses the image and language embeddings and the Frame-Scoring Transformer jointly attends to all frames to predict their relevance scores. During inference, the video summary is constructed by converting frame scores to shot scores and using Knapsack algorithm to select high scoring shots.

**Overview.** Our approach, CLIP-It, is outlined in Fig 2. Given a video, we extract $N$ frames denoted by $F_i$, $i \in [1, \ldots N]$. We formulate the video summarization task as a per-frame binary classification problem. We embed the frames using a pretrained network $f_{img}$. If a query is provided (in the form of a natural language string), we embed the query using a pretrained network $f_{txt}$. Alternatively, as seen in the figure, we use an off-the-shelf video captioning model to generate a dense video caption with $M$ sentences denoted by $C_j$, $j \in [1, \ldots M]$ and embed each sentence using the pretrained network $f_{txt}$. Next, we compute language attended image embeddings $I^*$ using learned Language-Guided Multi-head Attention $f_{img\_txt}^*$. Finally, we train a Frame-Scoring Transformer which assigns scores to each frame in the video (green indicating a high score and red indicating a low score). To construct the video summary during inference, we convert frame-level scores to shot-level scores and finally, use 0/1 knapsack algorithm to pick the key shots [43]. In the following, we describe the Language-Guided Attention and the Frame-Selection Transformer modules, followed by discussing the visual and language encoders.

**Language-Guided Attention.** We use Language-Guided Multi-head Attention to efficiently fuse information across the video and language modalities and to infer long-term dependencies across both. Using a single attention head does not suffice as our goal is to allow all captions to attend to all frames in the video. We modify the Multi-Head Attention described in Vaswani *et al.* [38] to take in inputs from both modalities. We set Query $Q$, Key $K$, and Value $V$ as follows,

$$Q = f_{img}(F_i), \text{ where } i \in [1, \ldots N],$$
$$K, V = f_{txt}(C_j), \text{ where } j \in [1, \ldots M],$$
$$\text{Language - Guided Attn.}(Q, K, V) = \text{Concat}(\text{head}_1, ..., \text{head}_h)W^O,$$
$$\text{where head}_i = \text{Attention}(QW_i^Q, KW_i^K, VW_i^V)$$
$$\text{and Attention}(Q, K, V) = \text{softmax}(\frac{QK^T}{\sqrt{d_k}})V$$

$W_i^Q$, $W_i^K$, and $W_i^V$ are learned parameter matrices and $d_k$ is the dimensions of $K$. The output of the Language-Guided Multi-Head Attention are the attended image embeddings, denoted as $F_i^{'}$.

**Frame-Scoring Transformer.** Finally, it is also important to ensure that we do not include redundant information, e.g., several key shots from the same event. To better model interactions across frames and contextualize them w.r.t. each other, we add a Frame-Scoring Transformer that takes image-text representations as input and outputs one score per frame. Based on the Transformer model [38], this module assigns relevance scores to the attended image embeddings $F_i^{'}$. We feed $F_i^{'}$ to the bottom of both the encoder and decoder stacks. Similar to [38], we use positional encoding to insert information

about the relative positions of the tokens in the sequence. We add positional encodings to the input embeddings at the bottom of the encoder and decoder stacks.

**Image Encoding.** We encode the image using a pre-trained network $f_{img}$. We experiment with the following networks: GoogleNet [37] (for a fair comparison to prior work), ResNet [8], and CLIP [29].

**Text Encoding.** We encode the user-defined query or the system generated dense caption using a pre-trained network $f_{txt}$. In this case we tried the CLIP (ViT and RN101) model. In case of generic video summarization, we generate dense video captions for the *whole video* in order to condition on language and incorporate added semantics into our video summarization pipeline. We use the Bi-Modal Transformer [11] dense video captioning model that generates a multi-sentence description given a video with audio. Since there are multiple sentences in the dense video caption, we first embed each sentence of the caption using the text encoder $f_{txt}$ described above. They are then concatenated and fused using a multi-layer perceptron (MLP).

## 3.1 Learning

We employ 3 loss functions (classification, diversity, and reconstruction) to train our model. The supervised setting uses all 3 and the unsupervised setting uses only diversity and reconstruction losses.

**Classification loss.** We use a weighted binary cross entropy loss for classifying each frame:

$$\mathcal{L}_c = -\frac{1}{N} \sum_{i=1}^{N} w^*[x_i^* \log(x_i)] + (1 - w^*)[(1 - x_i^*)\log(1 - x_i)], \tag{1}$$

where $x_i^*$ is the ground-truth label of the $i$-th frame and $N$ is the total number of frames in the video. $w^*$ is the weight assigned to the class $x_i^*$, which is set to $\frac{\#keyframes}{N}$ if $x_i^*$ is a keyframe and $1 - \frac{\#keyframes}{N}$ if $x_i^*$ is a background frame.

For the purpose of training without any supervision, we employ two additional losses that enforce diversity in the selected keyframes. We first select keyframes $X$ based on the scores assigned by the Frame-Scoring Transformer. We pass the output features of the transformer model for the selected keyframes through a decoder network consisting of $1 \times 1$ convolution layers to obtain reconstructed feature vectors for the selected keyframes such that each keyframe feature vector is of the same dimension as its corresponding input frame-level feature vector.

**Reconstruction Loss.** The reconstruction loss $\mathcal{L}_r$ is defined as the mean squared error between the reconstructed features and the original features corresponding to the selected keyframes, such that:

$$\mathcal{L}_r = \frac{1}{X} \sum_{i \in X} ||\mathbf{x}_i - \hat{\mathbf{y}}_i||_2, \tag{2}$$

where $\hat{\mathbf{y}}$ denotes the reconstructed features.

**Diversity Loss.** We employ a repelling regularizer [45] to enforce diversity among selected keyframes. Similar to [31, 30], we compute the diversity loss, $\mathcal{L}_d$, as the pairwise cosine similarity between the selected keyframes:

$$\mathcal{L}_d = \frac{1}{X(X-1)} \sum_{i \in X} \sum_{j \in X, j \neq i} \frac{\hat{\mathbf{y}}_i \cdot \hat{\mathbf{y}}_j}{||\hat{\mathbf{y}}_i||_2 \cdot ||\hat{\mathbf{y}}_j||_2}, \tag{3}$$

where $\hat{\mathbf{y}}_i$ and $\hat{\mathbf{y}}_j$ denote the reconstructed feature vectors of the $i$-th and $j$-th node.

The final loss function for supervised learning is then,

$$\mathcal{L}_{sup} = \alpha \cdot \mathcal{L}_c + \beta \cdot \mathcal{L}_d + \lambda \cdot \mathcal{L}_r, \tag{4}$$

where $\alpha$, $\beta$, and $\lambda$ control the trade-off between the three loss functions. We modify the loss function to extend CLIP-It to the unsupervised video summarization setting. We omit $\mathcal{L}_c$ since the groundtruth summary cannot be used for supervision and represent the final loss function for unsupervised learning as:

$$\mathcal{L}_{unsup} = \beta \cdot \mathcal{L}_d + \lambda \cdot \mathcal{L}_r, \tag{5}$$

where $\beta$ and $\lambda$ are balancing parameters to control the trade-off between the two terms. We include implementation details of our method in the Supp.

Table 1: **Supervised.** Comparing F1 Scores of our methods with supervised baselines on the SumMe [7] and TVSum [36] datasets using Standard, Augment, and Transfer data configurations.

| Method | SumMe | | | TVSum | | |
|---|---|---|---|---|---|---|
| | Standard | Augment | Transfer | Standard | Augment | Transfer |
| Zhang *et al.* (SumTransfer) [42] | 40.9 | 41.3 | 38.5 | - | - | - |
| Zhang *et al.* (LSTM) [43] | 38.6 | 42.9 | 41.8 | 54.7 | 59.6 | 58.7 |
| Mahasseni *et al.* (SUM-GAN$_{sup}$) [22] | 41.7 | 43.6 | - | 56.3 | 61.2 | - |
| Rochan *et al.* (SUM-FCN) [31] | 47.5 | 51.1 | 44.1 | 56.8 | 59.2 | 58.2 |
| Rochan *et al.* (SUM-DeepLab) [31] | 48.8 | 50.2 | 45.0 | 58.4 | 59.1 | 57.4 |
| Zhou *et al.* [47] | 42.1 | 43.9 | 42.6 | 58.1 | 59.8 | 58.9 |
| Zhang *et al.* [44] | - | 44.9 | - | - | 63.9 | - |
| Fajtl *et al.* [3] | 49.7 | 51.1 | - | 61.4 | 62.4 | - |
| Rochan *et al.* [30] | - | 48.0 | 41.6 | - | 56.1 | 55.7 |
| Chen *et al.* (V2TS) [1] | - | - | - | 62.1 | - | - |
| He *et al.* [9] | 47.2 | - | - | 59.4 | - | - |
| Park *et al.* (SumGraph) [26] | 51.4 | 52.9 | 48.7 | 63.9 | 65.8 | 60.5 |
| GoogleNet+bi-LSTM | 38.5 | 42.4 | 40.7 | 53.9 | 59.6 | 58.6 |
| ResNet+bi-LSTM | 39.4 | 44.0 | 42.6 | 55.0 | 61.0 | 59.9 |
| CLIP-Image+bi-LSTM | 41.1 | 45.9 | 44.9 | 56.8 | 63.7 | 61.6 |
| CLIP-Image+Video Caption+bi-LSTM | 41.2 | 46.1 | 45.5 | 57.1 | 64.3 | 62.4 |
| GoogleNet+Transformer | 51.6 | 53.5 | 49.4 | 64.2 | 66.3 | 61.3 |
| ResNet+Transformer | 52.8 | 54.9 | 50.3 | 65.0 | 67.5 | 62.8 |
| CLIP-Image+Transformer | 53.5 | 55.3 | 51.0 | 65.5 | 68.1 | 63.4 |
| **CLIP-It**: CLIP-Image+Video Caption+Transformer | **54.2** | **56.4** | **51.9** | **66.3** | **69.0** | **65.5** |

## 4 Experiments

In this section, we describe the experimental setup and evaluation of our method on two tasks: generic video summarization and query-focused video summarization.

### 4.1 Generic Video Summarization

Generic video summarization involves generating a single general-purpose summary to describe the input video. Note that while prior works only use visual cues from the video, our method also allows for video captions as an input feature. For a fair comparison, we include ablations of our method that do not use any language cues and only the visual features used in earlier works [43].

**Datasets.** We evaluate our approach on two standard video summarization datasets (TVSum [36] and SumMe [7]) and on the generic summaries for UT Egocentric videos [16] provided by the QFVS dataset [33]. TVSum [36] consists of 50 videos pertaining to 10 categories (how to videos, news, documentary, etc) with 5 videos from each category, typically 1-5 minutes in length. SumMe [7] consists of 25 videos capturing multiple events such as cooking and sports, and the lengths of the videos vary from 1 to 6 minutes. In addition to training on each dataset independently, we follow prior work and augment training data with 39 videos from the YouTube dataset [2] and 50 videos from the Open Video Project (OVP) dataset [24]. YouTube dataset consists of news, sports and cartoon videos. OVP dataset consists of multiple different genres including documentary videos. These datasets are diverse in nature and come with different types of annotations, frame-level scores for TVSum and shot-level scores for SumMe. They are integrated to create the ground-truth using the procedure in [43]. The UT Egocentric dataset consists of 4 videos captured from head-mounted cameras. Each video is about 3-5 hours long, captured in a natural, uncontrolled setting and contains a diverse set of events. The QFVS dataset [33] provides ground-truth generic summaries for these 4 videos. The summaries were constructed by dividing the video into shots and asking 3 users to select the relevant shots. The final ground-truth is an average of annotations from all users.

**Note.** All the datasets - YouTube [2], Open Video Project (OVP) dataset [24], TVSum [36], SumMe [7], and QFVS [32] were collected by the creators (cited) and consent for any personally identifiable information (PII) was ascertained by the authors where necessary.

**Data configuration for TVSum and SumMe.** Following previous works [43, 42, 31, 26], we evaluate our approach in three different data settings: Standard, Augment, and Transfer. In the Standard setting, the training and test splits are from the same dataset (i.e. either TVSum or SumMe). For SumMe we use available splits, and for TVSum we randomly select 20% of the videos for testing

Table 2: **Unsupervised.** Comparing F1 Scores of our methods with unsupervised baselines on the SumMe [7] and TVSum [36] datasets using Standard, Augment, and Transfer data configurations.

| Method | SumMe | | | TVSum | | |
|---|---|---|---|---|---|---|
| | Standard | Augment | Transfer | Standard | Augment | Transfer |
| Mahasseni *et al.* [22] | 39.1 | 43.4 | - | 51.7 | 59.5 | - |
| Yuan *et al.* [41] | 41.9 | - | - | 57.6 | - | - |
| Rochan *et al.* (SUM-FCN$_{unsup}$) [31] | 41.5 | - | 39.5 | 52.7 | - | - |
| Rochan *et al.* [30] | 47.5 | - | 41.6 | 55.6 | - | 55.7 |
| He *et al.* [9] | 46.0 | 47.0 | 44.5 | 58.5 | 58.9 | 57.8 |
| Park *et al.* (SumGraph) [26] | 49.8 | 52.1 | 47.0 | 59.3 | 61.2 | 57.6 |
| GoogleNet+bi-LSTM | 33.1 | 38.0 | 36.5 | 47.7 | 54.9 | 52.3 |
| ResNet+bi-LSTM | 34.5 | 40.1 | 39.6 | 51.0 | 56.2 | 53.8 |
| CLIP-Image+bi-LSTM | 35.7 | 41.0 | 41.4 | 52.8 | 58.7 | 56.0 |
| CLIP-Image+Video Caption+bi-LSTM | 36.9 | 42.4 | 42.5 | 53.5 | 59.4 | 57.6 |
| GoogleNet+Transformer | 50.0 | 52.7 | 47.6 | 59.9 | 62.1 | 58.4 |
| ResNet+Transformer | 50.8 | 53.9 | 49.3 | 61.1 | 63.0 | 59.9 |
| CLIP-Image+Transformer | 51.2 | 53.6 | 49.2 | 61.9 | 64.0 | 60.6 |
| **CLIP-It**: CLIP-Image+Video Caption+Transformer | **52.5** | **54.7** | **50.0** | **63.0** | **65.7** | **62.8** |

Table 3: Comparing F1 Scores for generic video summarization on the QFVS dataset.

| **Supervised** | Vid 1 | Vid 2 | Vid 3 | Vid 4 | Avg |
|---|---|---|---|---|---|
| SubMod [6] | 49.51 | 51.03 | 64.52 | 35.82 | 50.22 |
| QFVS [33] | 62.66 | 46.11 | 58.85 | 33.50 | 50.29 |
| CLIP-Image + bi-LSTM | 65.43 | 56.55 | 68.63 | 40.06 | 57.67 |
| ResNet + Transformer | 66.97 | 58.32 | 70.10 | 43.31 | 59.67 |
| CLIP-Image + Transformer | 70.8 | 61.67 | 72.43 | 47.48 | 63.11 |
| **CLIP-It** (Gen. Caption) | 74.13 | 63.44 | 75.86 | 50.23 | 65.92 |
| **CLIP-It** (GT Caption) | 84.98 | 71.26 | 82.55 | 61.46 | 75.06 |
| **Unsupervised** | Vid 1 | Vid 2 | Vid 3 | Vid 4 | Avg |
| Quasi [46] | 53.06 | 53.80 | 49.91 | 22.31 | 44.77 |
| CLIP-Image + Transformer | 65.44 | 57.21 | 65.10 | 41.63 | 57.35 |
| **CLIP-It** (Gen. Caption) | 67.02 | 59.48 | 66.43 | 44.19 | 59.28 |
| **CLIP-It** (GT Caption) | 73.90 | 66.83 | 75.44 | 52.31 | 67.12 |

and construct 5 different splits and report an average result on all 5 splits. For the Augment setting, the training set from one dataset (e.g., TVSum) is combined with all the data from the remaining three datasets (e.g., SumMe, OVP, and YouTube). This setting yields the best performing models due to the additional training data. The Transfer setting is the most challenging of the three. It involves training a model on three datasets and evaluating on the fourth unseen dataset.

**Quantitative Results.** We compare our method and its ablations to supervised video summarization baselines in Tab.1. We report F1 scores on the TVSum and SumMe datasets for all three data settings. Our full method, CLIP-It (CLIP-Image+Video Caption+Transformer), described in Sec. 3, outperforms state-of-the-art by a large margin on all three settings. Particularly, in the Transfer setting we outperform the previous state-of-the-art SumGraph [26] by 5% on TVSum and 3% on SumMe, indicating that our model is better than the baselines in generalizing to out-of-distribution data.

To prove the effectiveness of each component of our model, we include comparisons to different ablations. In CLIP-Image+Transformer, we ablate the Language-Guided Attention module and directly pass the CLIP-Image features as input to the transformer. As seen, the performance drops by 2% indicating that conditioning on CLIP language embeddings leads to better summaries. Substituting the Frame-Scoring Transformer with a Bidirectional LSTM in CLIP-Image+bi-LSTM and CLIP-Image+Video Caption+bi-LSTM again results in a performance drop, thus highlighting the need for the Transformer module in our model. For a fair comparison with the baselines, in GoogleNet+Transformer we use the same GoogleNet features provided by Zhang *et al.* [43] and *used by all the other baselines* and do not include language features. This method still outperforms Sum-

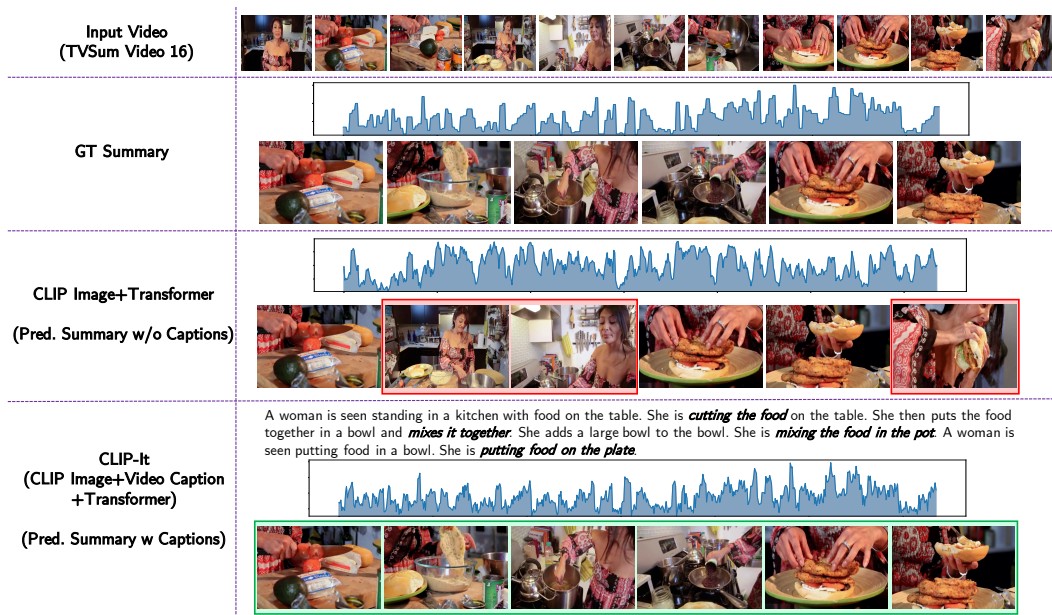

Figure 3: Comparison of ground-truth summary to results from CLIP-Image+Transformer and the full CLIP-It model (CLIP-Image+Video Caption+Transformer). The input is a recipe video. Without captions, the model assigns high scores to certain irrelevant frames such as scenes of the woman talking or eating which hurts the precision. With captions, the cross-attention mechanism ensures that frames with important actions and objects are assigned high scores.

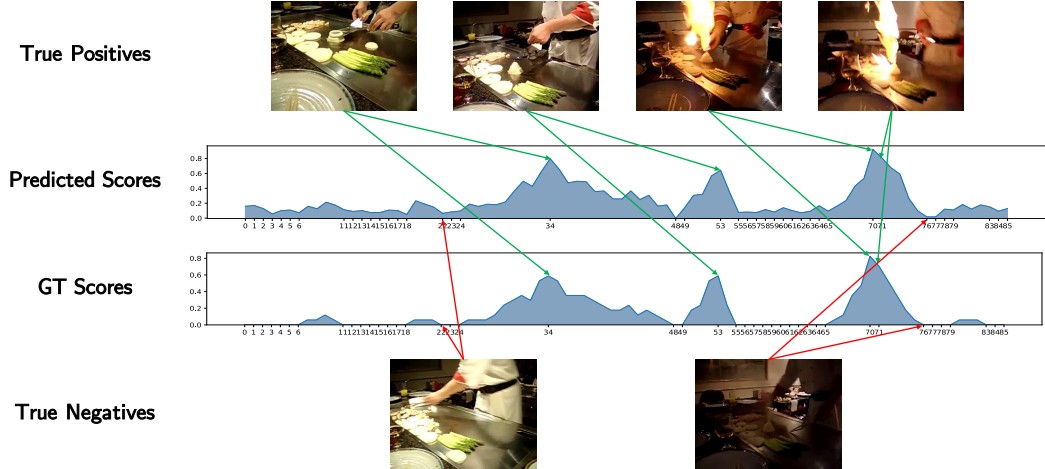

Figure 4: Qualitative result comparing the generic summary from CLIP-It with the ground-truth summary. The plots showing predicted and ground-truth frame-level scores are similar, indicating that frames that were given a high score in ground-truth were also assigned high scores by our model.

Graph [26]. Replacing the GoogleNet features with ResNet features results in a small performance improvement but CLIP-Image features prove to be most effective.

Tab. 2 compares F1 scores in the unsupervised setting to other unsupervised baselines. CLIP-It (CLIP-Image+Video Caption+Transformer), outperforms the best performing baseline on all settings on both datasets. We again observe large performance improvements in the Transfer setting and notice that overall our unsupervised method performs almost as well as our supervised counterpart. In CLIP-Image+Transformer we ablate the language component which causes a drop in performance, thus proving that language cues are useful in the unsupervised setting as well. Other ablations yield similar results reiterating the need for each component of our method. We also follow Otani *et al.* [25] and report results on rank based metrics, Kendall's $\tau$ [14] and Spearman's $\rho$ [49] correlation coefficients in the Supp.

Table 4: Results on UT Egocentric dataset [16]

| Method | F-Measure | Recall |
|---|---|---|
| **Supervised** | | |
| Submod-V+Both et al. [27] | 34.15 | 31.59 |
| CLIP Image + Transformer | 41.58 | 39.96 |
| **CLIP-It**: CLIP Image + Gen. Video Caption + Transformer | 44.70 | 43.28 |
| **CLIP-It**: CLIP Image + GT Video Caption + Transformer | 52.10 | 50.76 |
| **Unsupervised** | | |
| CLIP Image + Transformer | 39.22 | 37.46 |
| **CLIP-It**: CLIP Image + Gen. Video Caption + Transformer | 42.10 | 40.65 |
| **CLIP-It**: CLIP Image + GT Video Caption + Transformer | 49.98 | 47.91 |

Table 5: Results on TV Episodes dataset [16]

| Method | F-Measure | Recall |
|---|---|---|
| **Supervised** | | |
| Submod-V+Sem. Rep. et al. [27] | 40.90 | 37.02 |
| CLIP Image + Transformer | 47.82 | 46.02 |
| **CLIP-It**: CLIP Image + GT Video Caption + Transformer | 55.34 | 53.90 |
| **Unsupervised** | | |
| CLIP Image + Transformer | 45.77 | 44.01 |
| **CLIP-It**: CLIP Image + GT Video Caption + Transformer | 53.42 | 52.50 |

Tab. 3 shows F1 scores for the generic setting on the QFVS Dataset [33]. Following [33], we run four rounds of experiments leaving out one video for testing and one for validation, while keeping the remaining two for training. Our method, CLIP-It (Gen. Captions) outperforms both supervised and unsupervised baselines by a large margin on all four videos, and particularly on Video 4, which happens to be the most difficult for all methods. Adding captions helps significantly improve ( 2%) the summaries and outperforms the CLIP Image + Transformer baseline. To see how well our model would perform if we had perfect captions, we also show results by using the ground-truth captions obtained from VideoSet [40]. Replacing CLIP-Image features with ResNet features causes a drop in performance. Likewise, replacing the Transformer with a bi-LSTM also hurts performance.

We include results of our method (1) without captions (2) using generated captions (3) using the ground truth captions provided by VideoSet [40] for UT Egocentric and TV Episodes datasets in Tables 4 and 5. As ground truth, we obtain 15 summaries for each video using the same greedy n-gram matching and ordered subshot selection procedures as previous work [27]. We follow the same procedure as in prior work [40, 27] for creating and evaluating text summaries from video summaries. Our method outperforms [27] in both the supervised and unsupervised settings on both datasets.

**Qualitative Results.** We highlight the need to use language, specifically dense video captions, for constructing generic video summaries through a qualitative example in Fig. 3. The input is a video of a woman demonstrating how to make a chicken sandwich. The ground truth summary shows the scores computed by averaging the annotations from all users as in [43] and the keyframes that

Table 6: Comparing F1 Scores of different methods on the QFVS dataset.

| | Vid 1 | Vid 2 | Vid 3 | Vid 4 | **Avg** |
|---|---|---|---|---|---|
| SeqDPP [5] | 36.59 | 43.67 | 25.26 | 18.15 | 30.92 |
| SH-DPP [32] | 35.67 | 42.74 | 36.51 | 18.62 | 33.38 |
| QFVS [33] | 48.68 | 41.66 | 56.47 | 29.96 | 44.19 |
| CLIP-Image + Query + bi-LSTM | 54.47 | 48.59 | 62.81 | 38.64 | 51.13 |
| ResNet + Query + Transformer | 55.19 | 51.03 | 64.26 | 39.47 | 52.49 |
| **CLIP-It**: CLIP-Image + Query + Transformer | **57.13** | **53.60** | **66.08** | **41.41** | **54.55** |

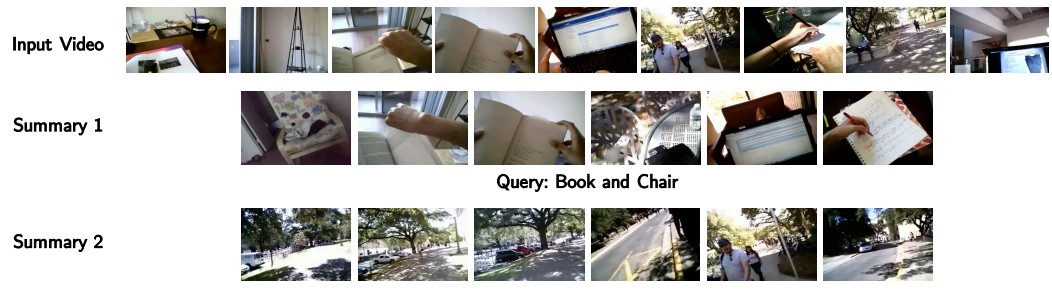

Figure 5: Result of our method on the QFVS dataset. The first row shows some frames from the 4 hour long input video. Given the query "book and chair", Summary 1 shows some frames selected by our method. Summary 2 shows frames for the query "Sun and Tree".

received high scores. Next we show the result from the baseline CLIP-Image+Transformer, which uses only visual features and *no language input*. The predicted scores show that the high scoring frames in the ground truth also receive a high score by our baseline, however a lot of irrelevant frames end up receiving a high score too. Thus, when the model finally picks the keyframes it ends up selecting frames where the person is talking or eating the sandwich (shown in red) which do not correspond to the key steps in the video. Adding language guidance via generated captions helps address this problem. The last row shows the captions generated by BMT [11]. The results shown are from our full CLIP-It model. The predicted scores are more similar to ground truth scores and the highest scoring keyframes are the same as the ground truth.

Fig. 4 shows compares summaries generated by our method to the ground truth summary on a cooking video from the SumMe dataset. Predicted scores are the frame-wise scores predicted by CLIP-It and GT Scores are the scores computed from user annotations [43]. The top row (True Positives) shows high scoring keyframes which are chosen by both our method and the ground truth, and the green arrows point to the assigned scores. As we see, they are clear, distinct and represent key actions in the recipe. The bottom row (True Negatives) shows frames which are assigned a low score (shown by the red arrows) and are not part of the final summaries (both GT and predicted). E.g., the first frame is irrelevant and corresponds to a segment between key steps, while the second frame has poor lighting and its hard to tell what is being done.

## 4.2 Query-Focused Video Summarization

In Query-Focused Video Summarization, the summarization process is conditioned on an input user query, thus, multiple summaries can be obtained for the same video using different input queries.

**Dataset.** We evaluate our method on the QFVS dataset based on the UT Egocentric videos described earlier. The dataset consists of 46 queries for each of the four videos and user-annotated video summaries for each query.

**Quantitative Results.** In Tab. 6, we compare F1 scores of our method to 3 baselines, SH-DPP [32], Seq-DPP [5], and QFVS [33]. Following [33], we run four rounds of experiments leaving out one video for testing and one for validation, while keeping the remaining two for training. Our full model achieves an avg F1 score of 54.55% outperforming the best baseline (44.19%) by 10%. We would like to point out that our method uses more recent image features compared to the baselines. At the same time, when switching from CLIP to ResNet image embedding, we still improve significantly over the baselines. We expect the improvement to also hold with weaker features (e.g., as demonstrated with GoogleNet results in Tab. 1, 2).

**Qualitative Results.** Fig. 5 shows a result on the QFVS dataset for UT Egocentric videos. The input is an egocentric video shot from a head-mounted camera and spans 4 hours. It consists of multiple events from a person's day, such as reading books, working on the laptop, walking in the streets, and so on. Summary 1 and 2 show results from our CLIP-It method when the input query is "Book and chair" and "Sun and tree" respectively. The frames shown are frames assigned high-scores by our method. As seen, given the same input video, different queries yield different summaries.

# 5 Discussion and Broader Impacts

We introduced CLIP-It, a unified language-guided framework for generic and query focused video summarization. Video summarization is a relevant problem with many use-cases, and our approach provides greater flexibility to the users, allowing them to guide summarization with open-ended natural language queries. We envision potential positive impact from improved user experience if adopted on video platforms such as YouTube. We rely on an off-the-shelf video captioning model [11] and a large-scale vision-language model (CLIP [29]) which may have encoded some inappropriate biases that could propagate to our model. Our visual inspection of the obtained summarization results did not raise any apparent concerns. However, practitioners who wish to use our approach should be mindful of the sources of bias we have outlined above depending on the specific use case they are addressing.

**Acknowledgements.** We thank Arun Mallya for very helpful discussions and feedback. We'd also like to thank Huijuan Xu for feedback on the draft. This work was supported in part by DoD including DARPA's XAI, LwLL, and SemaFor programs, as well as BAIR's industrial alliance programs.

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
