# OpenReview forum: "CLIP-It! Language-Guided Video Summarization"
_NeurIPS.cc/2021/Conference — NeurIPS 2021 Poster_

### Official Review · Reviewer_tjo5 · 2021-07-12

**Rating:** 6
**Confidence:** 5

**Summary:**

This paper proposes a new and uniform method for generic and query-focused video summarization. In addition to typical video input, the method also takes caption or user query as input, and thus learns a language-guided representation to help video summarization.

**Limitations And Societal Impact:**

Yes. The authors discussed the impacts of the proposed method in Sec. 5.

**Main Review:**

Strength:

- Applying transformers to model different modalities and improve video summarization.
- The proposed method outperforms state-of-the-art generic summarization methods on both benchmark dataset by 5% and 3%, respectively.

Weakness:

- The reconstruction loss: training the model to reconstruct keyframes is not very intuitive to the reviewer.  Any frame could be leant to reconstructed from the transformers, which does not help summarization, in particular for the unsupervised setting(s).
- The diversity loss only considers pairwise similarities among keyframes. This may be insufficient : as the major event may take a few frames to summarize, they are more similar to each other than a random frame (noise).
- There are manually annotated captions as ground truth for UT Egocentric dataset as shown in [A].
    - It would be more insightful to check the ground truth captions for video summarization.
    - [26] in this case could also be fair comparison.
- [26] can also use captions as input, the reviewer would encourage the authors to compare and report the results.
- The clarification of the paper could be improved:
    - Implementation: Are the weights on each loss summed up to 1? This is not claimed in the paper and chosen by heuristics.
    - Experiments: is the summary taking at most 15% of the whole video? This is very important for fair comparison. Also TVSum has a commonly used train/test split, as shown in [1, 2, 9, 25, 29, 42].
    - Typo: line 43, In case of generic; line 61, comprising of all; line 84, has lead to; line 258 Fig. 4 shows compares summaries; line 266, its hard to tell;

Additional comments:

There are many objects and actions in a video, but most can be  not relevant to the major activities  and considered as 'background'. It would be great to visualize the attention weights among video frames to captions/queries.

[A] Videoset: Video summary evaluation through text. Serena Yeung, Alireza Fathi, Li Fei-Fei. ArXiv 2014.

**Time Spent Reviewing:**

4.5 hrs

---

> ### Author Response · Authors · 2021-08-10
> **Response to Reviewer tjo5: Thanks for the feedback! We have addressed the comments below.**
>
> > Intuition behind reconstruction loss.
>
> Both reconstruction and diversity losses are computed by first selecting X keyframes based on the scores assigned by the transformer.  We will clarify this and update Eq 2 and Eq 3 to reflect this. The reconstruction loss minimizes the difference between the reconstructed feature representations of the keyframes in the predicted summary video and the input frame-level representation of those key frames in the input video. This acts as a regularizer by ensuring that the reconstructed features and the input features are similar. We will clarify this in the paper.
>
> > The diversity loss only considers pairwise similarities among keyframes. This may be insufficient : as the major event may take a few frames to summarize, they are more similar to each other than a random frame (noise).
>
> To compute diversity loss, we first select X keyframes based on the scores assigned by the transformer. The diversity loss ensures that the feature representation for the selected keyframes are different. The final summary video is obtained by converting keyframes to keyshots, following the protocol in prior works. While keyshots have many frames from the same event, keyframes are diverse and typically consist of a single frame from any event. The diversity loss is applied to the keyframes (not keyshots), and so pairwise similarity is sufficient. It ensures that the model picks keyframes that are visually different.
>
> > Using ground truth captions from VideoSet [1] and comparison to [2].
>
> Thanks for the suggestion, we have included these results. Please see note to all reviewers.
>
> > Implementation: Are the weights on each loss summed up to 1?
>
> Yes they sum to 1 and were chosen heuristically. We will clarify this and include details in the final version.
>
> > Summary length and train/test splits
>
> For a fair comparison we follow prior works and set the summary to be 15% of the whole video. We also use the same splits defined in previous works.
>
> > Typo: line 43, In case of generic; line 61, comprising of all; line 84, has lead to; line 258 Fig. 4 shows compares summaries; line 266, its hard to tell;
>
> Thanks, we will fix this.
>
> > Visualizing the attention weights among video frames to captions/queries.
>
> Thanks for the suggestion, we will include this in the final version.
>
> [1]  S. Yeung, A. Fathi, and L. Fei-Fei. Videoset: Video summary evaluation through text. arXiv:1406.5824, 2014
>
> [2] Bryan A Plummer, Matthew Brown, and Svetlana Lazebnik. Enhancing video summarization via vision language embedding. In CVPR, 2017

---

> > ### Comment · Reviewer_tjo5 · 2021-09-01
> > **Post-rebuttal Comments**
> >
> > Many thanks to the insightful reviews from all reviewers.
> >
> > After carefully reading the authors' rebuttal, I think it addressed my concerns and I appreciate the authors' efforts on extensive ablation studies, e.g. summary length, train/test splits and generated v.s. GT video captions. I'm keeping my original rating for this paper.

---

### Official Review · Reviewer_Uj81 · 2021-07-16

**Rating:** 6
**Confidence:** 4

**Summary:**

The paper describes a video summarization model that is trained to use either user provided query or automatically generated dense  captioning to select key-frames as summaries.

At the core of the approach is an attention mechanism that computes similarity between frame embeddings and text embeddings. Subsequently, each frame representation is replaced by a weighted-average (softmaxed) of the language embeddings; key and value embeddings are from the text model while query embeddings are from the image model. Finally, a score is assigned to each frame via a standard transformer based model. A post-processing step converts the key frame scores to key-shot level summaries such that only 15% of the videos are selected. The overall model is trained with a weighted Binary cross-entropy loss.


**Limitations And Societal Impact:**

Concerns are addressed.

**Main Review:**


Strength:
 - The idea of guiding visual summarization with language models is very nice. Intuitively, one would expect the caption/query of a video to represent the summary of visual concepts in it. I encourage the authors to explore the idea further.
- The paper is clearly written and well organized.
- The overall architecture (Clip Image + visual captioning + score transformers) is better than earlier approaches.

Weakness:
- Some important details about the model are not discussed (see main questions below)
- The motivation for guiding video summarization with language model, as far as my understanding goes, is to train systems that can learn visual representation in correlation with high-level concepts encoded in text-embeddings. The paper did not show, empirically, that the introduced attention led to better visual representation learning. If this is shown it would make the paper a strong submission.
  - I recommend to design an experiment where the system is trained with language-guided attention, and evaluated without generating captions or the attention mechanism. If such a system proves to be better, then it would be a strong validation for the very nice motivation.

Main questions:
- How are N frames selected? The sampling method is not discussed, however, the sampling could have huge difference which might lead to significant misalignment with respect to a query.
- In the case of the generated captions, are the videos subsampled for the caption generation? if so is the sampling for caption generation and visual embedding the same?
- The paper discusses using global attention (i.e., computing similarity between each frame and textual embedding). Does this not lead to a high-entropy distribution? I am referring to the distribution defined by the softmax of the similarity scores. This can become particular problematic for captions/queries that are very long.
- Do you think a (dense) video captioning model (like the one used here) along with the scoring transformer would suffice to achieve the results? In other words, to what extent does the language-guided attention architecture plays a role in getting the results?
  - Table 1. Shows the important components of the model, with respect to performance, are the quality of the embeddings and the transformer architecture. For instance, in case of [Clip Image + video caption + bi-LSTM] the approach is comparable or worse, in most cases, than related works.
- Can you say something about choosing the key and value embeddings to be from the text model rather than the video? Is this to avoid the the high-entropy problem in long sequences? ( see Question 3)

Significance:
- The work address an important problem in a research application that is early on its development.

**Time Spent Reviewing:**

4-5

---

> ### Author Response · Authors · 2021-08-10
> **Response to Reviewer Uj81: Thanks for the feedback! We have addressed the comments below.**
>
> > The motivation for guiding video summarization with language model, as far as my understanding goes, is to train systems that can learn visual representation in correlation with high-level concepts encoded in text-embeddings. The paper did not show, empirically, that the introduced attention led to better visual representation learning.
>
> Fig 3 in the paper includes a qualitative example where adding captions helps omit irrelevant frames; we'll add more such examples. We will also include visualizations of the attention weights on the frames from the captions/queries in the final version. We'd like to clarify that we *do not* claim that we learn better general visual representations. Instead, we aim to generate better summaries by efficiently fusing information across the video and language modalities and inferring long-term dependencies across both. Through empirical evaluation, in Tab 1 and Tab 2, we show that adding language-guided attention leads to performance improvement on video summarization. Please also see note to all reviewers where we include additional experiments to support the need for adding language to our pipeline.
>
> > I recommend designing an experiment where the system is trained with language-guided attention, and evaluated without generating captions or the attention mechanism.
>
> We have included an experiment where we replaced the language-guided attention block with a simple MLP, to show the need for cross-modal attention in our framework. Could you please clarify what it means to "train with language-guided attention but evaluate without generating captions or the attention mechanism"?
>
> > Frame extraction and sampling.
>
> Following prior works, we sample frames at 2 fps for obtaining the input image features. Similarly, for generating captions, we sample frames at 2 fps and feed them to BMT which returns a dense video description. We then sample 7 captions from the dense video description. To clarify our method, all the subsampled frames in the video and all the 7 captions are fed as input into the language-guided attention network. The network learns to assign importance weights to the image frames based on the captions.
>
> > Are frames sampled for caption generation? If so, is the sampling for caption generation and visual embedding the same?
>
> Yes and yes. We sample frames at 2 fps for both.
>
> > Does global attention not lead to a high-entropy distribution?
>
> We generate a dense video description for the video and uniformly sample 7 sentences from it. We then use CLIP to extract an embedding per sentence. The embeddings are fused using an MLP. The joint embedding is passed as both key and value pairs to the language-guided attention unit. There are a total of 7 textual embeddings attending to all the frames in the video. As such, we didn’t encounter any issues with high-entropy.
>
> > What is the role of the language-guided attention block?
>
> Thanks for the suggestion! To validate the effectiveness of the language-guided attention, we replaced this with a simple MLP that concatenates the Image and Text embeddings. Results on TV Sum and SumMe in a supervised setting are below. Without the attention block there’s a ~4% drop in performance
>
> | Method |  | SumMe |  |  | TVSum | |
> | --- | :---: | :---: | :---: | :---: | :---: | :---:
> |  | Standard | Augment | Transfer | Standard | Augment | Transfer
> | CLIP Image+Video Caption(MLP)+Transformer | 50.6 | 51.08 | 48.1 | 63.0 | 65.8 | 61.4
> | CLIP-It: CLIP-Image+Video Caption(Attn)+Transformer  | 54.2 | 56.4 | 51.9 | 66.3 | 69.0 | 65.5
>
> > Why is there no improvement when adding captions to bi-LSTM?
>
> While adding captions is helpful as seen above, it is the combination of CLIP features with the language-guided attention framework and the transformer architecture that leads to the best performance. We hypothesize that the low increase in performance when adding captions to the bi-LSTM architecture is due to the inability of the bi-LSTM to attend to a large temporal window, rather than an issue with the captions.
>
> > Why are key and value embeddings from the text model rather than the video? Is this to avoid high-entropy in long sequences?
>
> In the Multiheaded Attention block in “Attention is All You Need”, the dimensions of the output sequence match that of the query. For this reason, the query is set to be the embeddings from the video model and the key and value are chosen from the text model. It is correct that this also avoids the high entropy problem as stated in the previous response.

---

> ### Comment · Reviewer_Uj81 · 2021-08-25
> **Comments after rebuttal.**
>
> Thank you for the response. Some of the answers did clarify my concerns.
>
> My main concern was that the language-guided attention, which is the main and interesting part of the paper, was not shown as effective mechanism to learn visual representations, i.e., the projection maps of the attention mechanism for the visual features, influenced by the text-embeddings.  As the authors answered, that is not their claim. Consequently, it makes the work less interesting but still within the scope of its claim.
>
> As a result, I would suggest to update the manuscript by emphasizing and including the details of how the generated/given captions are sampled into 7 sentences. Since, that is effectively where the summaries of the video are done, and the network might simply be learning to match them with visual features.  The better the captions the better the summaries are, see tables above.
>
> If such updates are included into the manuscript, I am leaning towards accept pending the final suggestion of the other reviewers.

---

> > ### Author Response · Authors · 2021-08-25
> > **Response to comments after rebuttal.**
> >
> > We're glad we were able to clarify your concerns. As per your suggestions, we will update L140-L142 in the manuscript to be, "Alternatively, we use an off-the-shelf video captioning model to generate a dense video caption, as seen in the figure. We then uniformly sample M sentences from the description denoted by $C_j, j \in [1, . . . M]$ such that they span the entire duration of the video, i.e. if the description contains 14 sentences and we need to sample 7, then we pick every other sentence. We then embed each sentence using the pre-trained network $f_{txt}$".  In Sec 2. Implementation Details in the Supplementary, we will add this, "Following prior works, we sample frames at 2 fps for obtaining the input image features. Similarly, for generating captions, we sample frames at 2 fps and feed them to BMT which returns a dense video description. We set $M=7$ and then sample 7 captions from the dense video description. To clarify our method, all the subsampled frames in the video and all the 7 captions are fed as input into the language-guided attention network. The network learns to assign importance weights to the image frames based on the captions."  To show that better captions lead to better summaries, we will include all the additional experimental results incorporated in the rebuttal in the manuscript Sec 4. Experiments.
> >
> > Please note that since we cannot update the manuscript during the review process, we will make these changes in the final version.

---

> > ### Author Response · Authors · 2021-09-02
> > **Response to comments after rebuttal**
> >
> > Dear reviewer,
> >
> > We have addressed your concerns in the response below and we hope you will update your rating accordingly.
> >
> > Thanks -
> > Authors 1853

---

### Official Review · Reviewer_X5gK · 2021-07-18

**Rating:** 7
**Confidence:** 4

**Summary:**

This paper unified the generic video summarization method and the query-based video summarization method. Specifically, the paper proposed a language guided video summarzation approach that takes both video frames and text as input and summarizes the video frames under the influence of the text. For generic video summarization, the model relies on a video captioning method to provide the text guidance. The proposed approach achieves superior performance on both generic video summarization datasets and the query-based video summarization datasets.

**Limitations And Societal Impact:**

Yes

**Main Review:**

Method:

1. I have concerns with the generic video summarzation part. The proposed approach relies on video captioning method to provide a text that describe the event in the video. This seems pretty straightforward conceptually. However, in practical, it relies on a very good video captioning system.

  1.a The videos in summarzation datasets are very long. I wonder how many sentences are generated by the video captioning method? How to generate a caption for a day-long video? Did you sub-sample the frames?

  1.b The genres of videos in the summarzation dataset might be differ to the videos for training the video captioning model. How to handle the domain gap?

  1.c It seems the main reasons for the improvement in Table 1 and Table 2 are 1. Using Transformer. 2. Using better image feature (CLIP). The improvement brought by the caption is very limited (0.1 in Table1 under the setting of CLIP-Image + bi-LSTM. 0.7 in Table 1 under the setting of CLIP-Image + Transformer). The caption part only brought marginally improvement. It seems suggest the Caption is not useful?

  1.d If the caption is helpful for video summarization.  I wonder what property would the caption need to improve the performance? Does the caption need to be concise or complicated? There are multiple off-the-shelf video captioning system. Why would the paper use the approach proposed by Iashin et. al?

  1.e It would be better to show the example of captions in SumMe and TVSum.

---

Rating:

My main concerns are with the captioning part and the ablation study in Table 1. Conceptually I think the proposed approach makes sense and the direction is very interesting. However, I have doubt on whether the experiments support the claim. I would happy to raise the score to 7 if the concerns are answered.


**Time Spent Reviewing:**

3

---

> ### Author Response · Authors · 2021-08-10
> **Response to Reviewer X5gK: Thanks for the feedback! We have addressed the comments below.**
>
> > Concerns with generic video summarization and the need for a good captioning system.
>
> In the paper we included an ablation of our method, CLIP Image+Transformer, that doesn’t use captions. As seen in Tables 1, 2 and 3, the CLIP Image+Transformer ablation still outperforms SOTA. As automatic captioning systems improve, so will the performance of our summarization method. To verify this, we include results on UT Egocentric Videos and TV Episodes datasets with ground truth captions provided by the VideoSet dataset. Please see our note to all reviewers. We’ve also included results by using generated captions on the QFVS dataset (generic summarization) below. We sampled 15 captions for each video since the videos are significantly longer. We trained CLIP-It: CLIP-Image+Video Caption+Transformer using these captions and include results in the table below. Adding generated captions helps significantly improve (~2%) the summaries and outperforms the CLIP Image + Transformer baseline. Further, using ground-truth captions from VideoSet[1] we achieve ~10% improvement.
>
> | Supervised   | Vid 1 | Vid 2 | Vid 3 | Vid 4 | Avg
> | --- | :---: | :---: | :---: | :---: | :---:
> | CLIP Image + Transformer | 70.86 | 61.67 | 72.43 | 47.48 | 63.11
> | CLIP-It: CLIP Image + Gen. Video Caption + Transformer | 74.13 | 63.44 | 75.86 | 50.23 |  65.92
> | CLIP-It: CLIP Image + GT Video Caption + Transformer | 84.98  | 71.26  | 82.55  | 61.46 | 75.06
>
> | Unsupervised | Vid 1 | Vid 2 | Vid 3 | Vid 4 | Avg
> | --- | :---: | :---: | :---: | :---: | :---:
> | CLIP Image + Transformer | 65.44 | 57.21 | 65.10 | 41.63 | 57.35
> | CLIP-It: CLIP Image + Gen. Video Caption + Transformer | 67.02 |  59.48 | 66.43 | 44.19 |  59.28
> | CLIP-It: CLIP Image + GT Video Caption + Transformer | 73.90 | 66.83 | 75.44 | 52.31 | 67.12
>
> > Frame extraction and caption generation for long videos. How many sentences are sampled?
>
> For the TVSum and SumMe datasets, as described in the Supp Sec. A2 (Text Encoding), we first generate dense video descriptions using BMT [1] by sampling frames from the input video at 2 fps. For a 2-3 min video BMT generates ~10-15 sentences. Next, we uniformly sample 7 sentences from the dense description corresponding to different video segments over time. Each sentence is then encoded using CLIP text encoder and the 7 embeddings are concatenated to obtain a feature vector. This is passed through a linear layer to obtain the input text embedding. Heuristically, we found that sampling 7 captions worked best for TVSum and SumMe datasets where the average duration of the videos is ~2 mins. For generic summarization on the QFVS dataset (day long videos) reported above, the frames are extracted at 2 FPS and pass this through the BMT pipeline. This generates roughly 20 sentences and we then sampled 15 captions for each video since the videos are significantly longer. We will include these details in the final version.
>
> > Handling domain gap between summarization and captioning datasets.
>
> BMT [1] is good at generalizing to out-of-distribution data and achieves high scores on the open-domain Activity Net captions dataset. We found that it works well for our domain and generates reasonable captions as seen in Fig 3.
>
> > Does captioning improve performance? Why not with bi-LSTM?
>
> Please see note to all reviewers. The results of our method on TV Episodes and UT Egocentric using ground-truth captions shows that our language-guided attention mechanism leverages the information in the captions to generate better summaries. Additionally, we also include results of CLIP-It on the QFVS dataset above. As seen, it outperforms our baseline that doesn’t use captioning by ~2%. Fig 3 in the paper is another example where using captions helps the model pick scenes with relevant actions and objects as denoted in the captions. While adding captions is helpful, it is the combination of CLIP features with the language-guided attention framework and the Transformer architecture that leads to the overall best performance. We hypothesize that the low increase in performance when adding captions to the bi-LSTM architecture is due to the inability of the bi-LSTM to attend to a large temporal window, rather than an issue with the captions.
>
> > What kind of captions lead to better summaries (concise/complicated)? Why choose BMT for caption generation?
>
> We found that captions containing around 7-10 sentences that describe just the main sequence of events in the video without getting into the details are best for video summarization. Thus, they need to be concise and not complicated. We chose BMT as the pre-trained model and code was available and easy to run. It performs competitively on the open-domain Activity Net captions dataset (https://paperswithcode.com/sota/dense-video-captioning-on-activitynet), and generalizes well to out of distribution data. Prior methods relied on a legacy feature extraction code which was difficult to reproduce. BMT also generalized best to the videos in our dataset.
>
> > Example of generated captions.
>
> Fig 3 shows an example of the generated caption for a video in the TV Sum dataset. We will include more examples in the final version.
>
> [1] Vladimir Iashin and Esa Rahtu. A better use of audio-visual cues: Dense video captioning with bi-modal transformer. BMVC, 2020.

---

> ### Comment · Reviewer_X5gK · 2021-09-01
> **Comments after rebuttal**
>
> During my first round review, my major concerns were 1. the effectiveness of proposed caption-guided video summarization method. and 2. If video caption is helpful for the video summarization approach, what kind of property would the caption require.
>
> In the rebuttal, the author resolved both of my concerns.
>
> 1. For the effectiveness part, the author conducted more experiments on QFVS shows the effectiveness of caption-guided summarization. The improvement with both generated and ground-truth caption are pretty significant in both supervised and unsupervised setting. I think this experiments verify the effectiveness of the proposed approach.
>
> 2. Given the improvement brought by the caption-guided method, the model's performance should depend on the quality of generated caption. The author leverages a generic video captioning approach to solve this issue. It would pretty interesting to see the caption-guided summarization performance based on different captioning approaches.
>
> 3. 'We found that captions containing around 7-10 sentences that describe just the main sequence of events in the video without getting into the details are best for video summarization.' This is a very interesting observation that would inspire multiple new follow-up approaches. I think this observation should be included in the manuscript.
>
> In general, I think the author's rebuttal resolved my concerns. The proposed approach and direction are very interesting. It should inspire a bunch of follow-up works. I think this paper should be accepted. I would be glad to upgrade my rating to 7, if the author updated the manuscript by including the updated QFVS results.

---

> > ### Author Response · Authors · 2021-09-01
> > **Response to Comments after rebuttal**
> >
> > Thank you for your feedback! We are glad we were able to address your concerns. Please note that since we cannot update the manuscript during the review process, we will make these changes in the final version:
> >
> > 1. We will include the additional results using ground truth QFVS captions in Section 4 Experiments.
> > 2. We will add the line, "We found that captions containing around 7-10 sentences that describe just the main sequence of events in the video without getting into the details are best for video summarization."  to the Section 2 Implementation Details in the Supplementary.
> > 3. Additionally, as per your suggestion, we will include caption-guided summarization performance based for different video captioning methods.

---

### Author Response · Authors · 2021-08-10
**Note to all reviewers**

We thank the reviewers for their constructive comments!
We are excited that the reviewers recognized that our paper "is clearly written and well organized" [Uj81], "the idea of guiding visual summarization with language models is very nice" [Uj81], "proposed approach achieves superior performance on both generic video summarization datasets and the query-based video summarization datasets [X5gK, tjo5].
First, we address some of the more general concerns and highlight the new experiments. Further, we address individual reviewers’ concerns.

### 1. Benefit of captions
We include additional experiments that support our claim that language serves as an effective prior for event saliency and our model leverages information in captions to generate better visual summaries. As recommended by reviewer tj05, we include results of our method using the ground truth captions provided by VideoSet for UT Egocentric and TV Episodes datasets and also include comparisons to [2]. As ground truth, we obtain 15 summaries for each video using the same greedy n-gram matching and ordered subshot selection procedures as previous work [2]. We also report results on the unsupervised variant of our method that doesn’t require ground truth summaries. During inference, we follow the same procedure as in prior work [1,2] for creating and evaluating text summaries from video summaries and we report the recall and f-measure on each dataset on the ROUGE-SU score. We show results of our method (1) without captions (2) using generated captions (3) using GT captions. In all three scenarios, our method outperforms [2] in both the supervised and unsupervised settings on both datasets.

#### **UT Egocentric**

| Method | F-Measure | Recall
| --- | :---: | :---:
| **Supervised**  | |
| Submod-V+Both [2] | 34.15 | 31.59
| CLIP Image + Transformer | 41.58 | 39.96
| CLIP-It: CLIP Image + Gen. Video Caption + Transformer | 44.70 | 43.28
| CLIP-It: CLIP Image + GT Video Caption + Transformer | 52.10 | 50.76
| **Unsupervised**  | |
| CLIP Image + Transformer | 39.22 | 37.46
| CLIP-It: CLIP Image + Gen. Video Caption + Transformer | 42.10 | 40.65
| CLIP-It: CLIP Image + GT Video Caption + Transformer | 49.98 | 47.91


#### **TV Episodes**

| Method | F-Measure | Recall
| --- | :---: | :---:
| **Supervised**  | |
| Submod-V+Sem. Rep. [2] | 40.90 | 37.02
| CLIP Image + Transformer | 47.82 | 46.02
| CLIP-It: CLIP Image + GT Video Caption + Transformer | 55.34 | 53.90
| **UnSupervised** | |
| CLIP Image + Transformer | 45.77 | 44.01
| CLIP-It: CLIP Image + GT Video Caption + Transformer | 53.42 |  52.50

### 2. CLIP-It on QFVS Dataset

As recommended by reviewer X5gK, we added results of CLIP-It on the QFVS dataset (generic summarization) by generating captions for the day-long videos. Adding captions helps significantly improve (~2%) the summaries and outperforms the CLIP Image + Transformer baseline. To see how well our model would perform if we had perfect captions, we also show results by using the ground-truth captions obtained from VideoSet [1].

| Supervised   | Vid 1 | Vid 2 | Vid 3 | Vid 4 | Avg
| --- | :---: | :---: | :---: | :---: | :---:
| CLIP Image + Transformer | 70.86 | 61.67 | 72.43 | 47.48 | 63.11
| CLIP-It: CLIP Image + Gen. Video Caption + Transformer | 74.13 | 63.44 | 75.86 | 50.23 |  65.92
| CLIP-It: CLIP Image + GT Video Caption + Transformer | 84.98  | 71.26  | 82.55  | 61.46 | 75.06

| Unsupervised | Vid 1 | Vid 2 | Vid 3 | Vid 4 | Avg
| --- | :---: | :---: | :---: | :---: | :---:
| CLIP Image + Transformer | 65.44 | 57.21 | 65.10 | 41.63 | 57.35
| CLIP-It: CLIP Image + Gen. Video Caption + Transformer | 67.02 |  59.48 | 66.43 | 44.19 |  59.28
| CLIP-It: CLIP Image + GT Video Caption + Transformer | 73.90 | 66.83 | 75.44 | 52.31 | 67.12

### 3. Replacing Language-Guided Attention with MLP

Reviewer Uj81 suggested removing the language-guided attention block to verify its effectiveness. In the ablation below, we replace this block with a basic MLP. As seen, the performance drops by ~4% thus proving the need for the multi-headed attention between the two modalities.

| Method |  | SumMe |  |  | TVSum | |
| --- | :---: | :---: | :---: | :---: | :---: | :---:
|  | Standard | Augment | Transfer | Standard | Augment | Transfer
| CLIP Image+Video Caption(MLP)+Transformer | 50.6 | 51.08 | 48.1 | 63.0 | 65.8 | 61.4
| CLIP-It: CLIP-Image+Video Caption(Attn)+Transformer  | 54.2 | 56.4 | 51.9 | 66.3 | 69.0 | 65.5


[1]  S. Yeung, A. Fathi, and L. Fei-Fei. Videoset: Video summary evaluation through text. arXiv:1406.5824, 2014

[2] Bryan A Plummer, Matthew Brown, and Svetlana Lazebnik. Enhancing video summarization via vision language embedding. In CVPR, 2017

---

### Decision · Program_Chairs · 2021-09-27

**Decision:**

Accept (Poster)

**Comment:**

This submission received the following final ratings: 6, 6, 7.

Reviewer X5gK had initially expressed concerns about the effectiveness of the approach and the value of captions for summarization. The responses on these points provided by the authors convinced the Reviewer to raise the rating to 7.

Reviewer Uj81 gives a rating of 6, but recommends adding details about the sampling of the generated captions to the paper.

Reviewer tjo5 appreciates the clarifications given in the author response and confirms the original rating.

The ACs agree with the recommendation of acceptance.